# Experimental Study on the Effects of In Situ Stress on the Initiation and Propagation of Cracks during Hard Rock Blasting

**Guangliang Yan** [1,2], **Qibo Yang** [1,2], **Fengpeng Zhang** [1,2,*], **Qiqi Hao** [1,2], **Xiulong Wang** [1,2] and **Haonan Wang** [1,2]

1    Key Laboratory of Ministry of Education on Safe Mining of Deep Metal Mines, Northeastern University, Shenyang 110819, China; yanguangliang@stumail.neu.edu.cn (G.Y.); yangqibo@stumail.neu.edu.cn (Q.Y.); haoqiqi@stumail.neu.edu.cn (Q.H.); 1901041@stu.neu.edu.cn (X.W.); wanghaonan@stumail.neu.edu.cn (H.W.)
2    College of Resources and Civil Engineering, Northeastern University, Shenyang 110819, China
*    Correspondence: zhangfengpeng@mail.neu.edu.cn; Tel.: +86-024-8368-1958

**Featured Application: The specific application or a potential application of this study is related to rock engineering of deep high in situ stress.**

**Abstract:** In situ stress is one of the most important factors affecting rock dynamic fractures during blasting excavation of deep rock mass that generally is hard rock. In this research, crater blasting experiments on hard rock under different uniaxial static stresses were conducted to investigate the initiation and propagation process of crack networks that were induced by coupled dynamic and static loads. Furthermore, the effects of anisotropic static stress fields on the initiation and propagation of crack networks during hard rock blasting, and the crack network morphological characteristics were analyzed and elucidated. The experimental results showed that the static stress field changed the process of crack network initiation and propagation during hard rock blasting, and then control the crack network morphology. Under uniaxial static stress, the crack network was elliptical with the long axis parallel to the static stress. In addition, the larger the anisotropic static stress is, the more obvious the elliptical morphology of the crack network. Moreover, the static stress lead to the delay of crack formation which indicates that the delay time during millisecond blasting excavation of deep rock mass should be adjusted appropriately according to the in situ stress. A stress-strength ratio (SSR) of 0.15 is the threshold value where static stress may have a significant effect on the initiation and propagation of a crack network. Meanwhile, the strain field prior to crack initiation during rock blasting controlled the morphological characteristics of the crack network. Finally, the mechanism of static stress affecting propagation and morphology of crack network was revealed theoretically.

**Keywords:** crater blasting; in situ stress; crack initiation and propagation; dynamic fracture

## 1. Introduction

The development and utilization of deep underground mineral resources and underground space will become an inescapable trend in the future [1–5]. As the deep rock mass consists generally of hard rock, drilling and blasting methods are still widely applied in deep rock engineering. However, there are few methodologies that have been specifically developed for underground production blasting, and the few that have been developed do not adequately consider in situ stress [6–8].

As the depth increases, the in situ stress also increases gradually [9,10]. The result of a geo-stress survey in the Jinping II Hydropower Station, China, showed that the geo-stress increases with an increasing burial depth and that this geo-stress changes from a horizontal stress state to a vertical stress state with the increasing of the burial depth from 600 to 3000 m. The maximum in situ stress that was obtained from the survey reached up to 42.11 MPa [11,12]. Owing to the deep in situ stress, the blasting excavation of deep rock mass is completed under the coupling of dynamic and static loads [13–15]. Unfortunately,

in situ stress causes many problems for rock blasting such as boulder yield, serious over and under excavation, and rock burst [11,12,16–18]. Therefore, rock blasting where the stresses are high has attracted considerable attention from researchers around the world [19,20].

Previous studies have shown that static stress has an important effect on the mechanical properties of rock, such as compressive strength, tensile strength, Poisson's ratio, and Young's modulus [21–23]; the static stress controls the evolution of the strain field in rock mass during rock blasting. The strain fields under uniaxial and biaxial static stress are significantly different with those that are under no static stress [24,25]. Furthermore, the static stress controls the dynamic fracture behavior of the rock.

Under uniaxial static compressive stress, the cracks that are produced by explosion dynamic load tend to propagate along the axis of the direction of the static compressive stress [26–28]. In addition, Yi et al. [29] and Peng et al. [30] found, by using numerical experiments, that the higher the uniaxial static compressive stress, the longer the distance of crack propagation. However, under a triaxial static compressive stress, the number of cracks that are induced by dynamic loads will greatly reduce, as will the degree of damage to the rock [31,32]. Additionally, under the same dynamic loads, the greater the static stress that is perpendicular to the dynamic loads, the narrower the cracks of the rock [33]. Many studies that have investigated rock blasting tend to support the view that stress waves are responsible for the development of a damage zone in the rock mass, while the explosion gases are important in separating the crack pattern, and in throwing the fragments [34–36]. However, the static stress makes the failure process of rock under a blasting load more complex. Therefore, the dynamic fracture mechanism of rocks under static stress is different from those without static stress.

The magnitude of static stress also plays a significant role in the fracture mode of rock. Under a low static stress, dynamic load plays a major role in rock dynamic fracture, resulting in a tensile splitting failure in an intact specimen. Under a medium axial static stress, both of the static and dynamic loads play a major role in rock dynamic fractures, resulting in a composite tensile-shear failure in an intact specimen. Under a high axial static stress, static load plays a major role in rock dynamic fractures where the shear failure occurs in intact specimens [37]. Zhao et al. [38] found that in situ stress enhances the compression effect and weakens the tension effect in the radial direction of a borehole. They also found that, with increasing in situ stress, the radial crack zone presents a declining trend. From the previous studies on rock failure under the coupled dynamic and static loads, it can be concluded that the magnitude of the static compressive stress, dimensions, and the angle between the dynamic and static loads are important factors affecting the initiation and propagation of rock cracks. The anisotropic static stress leads to the spatial anisotropy of the distribution of rock blasting cracks, which eventually leads to a boulder yield, over or under excavation during rock blasting engineering.

Rock blasting failure is a process that contains the initiation and propagation of multiple cracks, during which the cracks interact with each other. It is necessary to comprehensively investigate the effects of hydrostatic stress on the formation and propagation of a crack network during rock blasting. Livingston crater blasting theory [39,40] is the theoretical basis for the design of rock blasting excavation. Therefore, Livingston crater blasting experiments on hard rock under static stress conditions can help to get insight into the effects of static stress on rock dynamic fracture behavior.

In this paper, a self-developed experimental apparatus for high stress hard rock blasting was used to conduct crater blasting experiments on green sandstone specimens under various uniaxial static stresses to study the initiation and propagation of the crack networks. Furthermore, the effect of static stress on the morphological characteristics of crack network during hard rock blasting were revealed, which is helpful to deep underground rock engineering such as mining and tunnel engineering.

## 2. Experiments

### 2.1. Experimental Apparatus

The experimental apparatus mainly consisted of an electrical explosion experimental system, a static loading system, a measuring system, and a control system [41,42]. The photos of the experimental apparatus are shown in Figure 1.

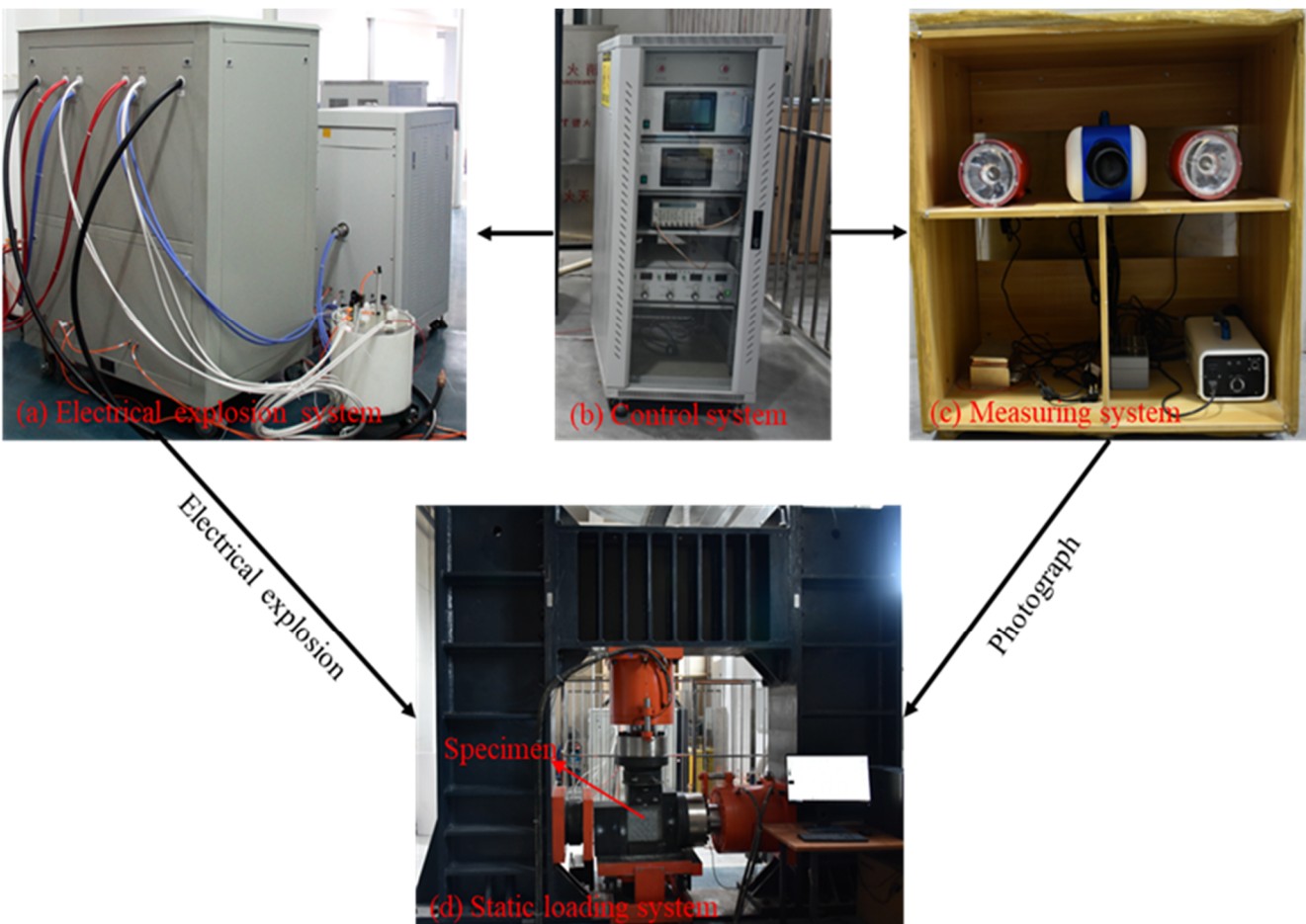

**Figure 1.** Experimental apparatus. (**a**) Electrical explosion experimental system; (**b**) control system; (**c**) measuring system; (**d**) static loading system.

The electrical explosion experimental system used a metal wire electrical explosion to simulate an explosive explosion and to apply the explosive dynamic load to the rock. The main parameters of the electrical explosion system were: a total capacitance of 4 μF, a rated charging voltage of 60 kV, and an energy storage capacity of 7.2 kJ.

The static loading system, with a maximum load capacity of 5000 kN, was a biaxial loading press, which applied a static stress field to specimens by force-controlled or displacement-controlled methods. A static load could be applied on specimens of various sizes, with the maximum specimen size being 500 mm × 500 mm × 250 mm.

The measuring system was an ultra-high-speed camera which was used to collect the failure process of the specimen during crater blasting. The main technical parameters of the camera were: a resolution of 924 (H) × 768 (V) and a maximum filming speed of 5 million frame/s with 180 frames per shot.

The control system, with a precision of 5 ps and a time delay in the range of 1 ns to 999 s, was a delay synchronization controller, whose function was to control the synchronization or delay start of the electrical explosion system and the measuring system. Thus, coordination between the blasting and the measuring system could be realized.

### 2.2. Specimen

The green sandstone that was used in this research was obtained from a quarry in Yunnan province, China. Table 1 lists the quasi-static properties of the green sandstone, which were obtained by using the ISRM recommended methods [43,44]. The samples were prepared as rectangular prismatic specimens with a size of 300 mm × 300 mm × 150 mm, with an aspect ratio of 2:1, as shown in Figure 2. Every specimen had a blast hole with a diameter of 11 mm and a length of 130 mm at its centre. The resistance line of each specimen was 20 mm.

**Table 1.** Quasi-static properties of the green sandstone.

| Density kg/m$^3$ | Young's Modulus GPa | Poisson's Ratio | Wave Velocity m/s | Compressive Strength MPa | Tensile Strength MPa |
|---|---|---|---|---|---|
| 2472 | 27.9 | 0.22 | 5028 | 80.7 | 4.2 |

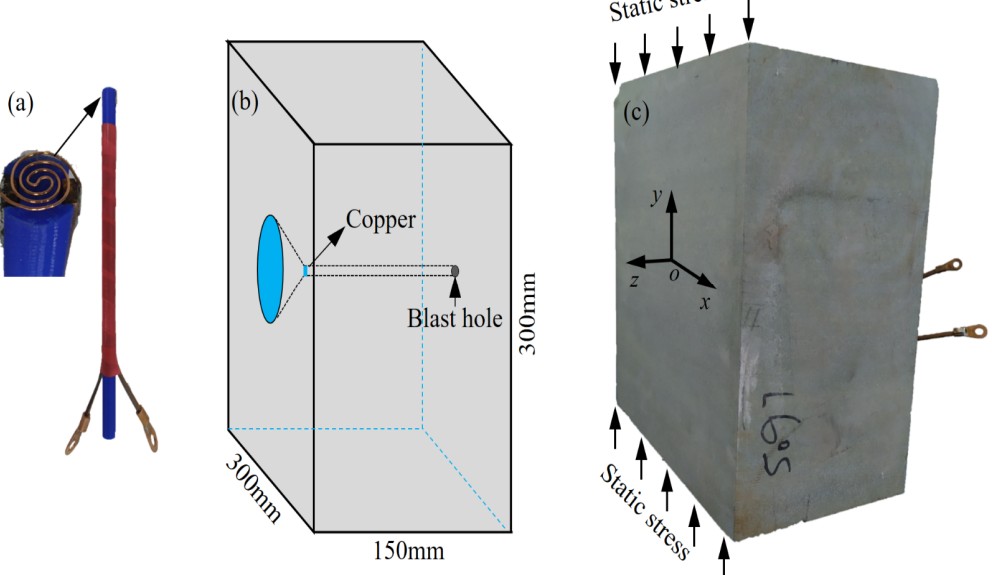

**Figure 2.** Schematic diagram and photo of specimen. (**a**) Electrode; (**b**) schematic diagram; (**c**) specimen.

In this study, the electrode, with a diameter of 10 mm, was used to simulate the explosion of the explosives, and then apply a dynamic load on the specimen, as shown in Figure 2a. The electrode was insulated by an insulating rod so as to avoid energy loss due to direct discharge during the test. The intermediate insulating rod was made of polyethylene by 3D printing. Before the wire explosion in the blast hole, an electrode and a coiled copper wire were inserted into the blast hole, and an acrylic adhesive was injected into the blast hole as the stemming material.

### 2.3. Experimental Scheme

In this study, the experiments were divided into 6 groups, and the uniaxial in situ stresses of 0, 8, 12, 16, 24, and 32 MPa were applied, respectively, corresponding to the stress-strength ratios (SSR) of 0, 0.1, 0.15, 0.2, 0.3, and 0.4. To ensure that the in situ stress remained unchanged during testing, the force-controlled loading method was applied. The charging voltage was 50 kV, the total energy was 5.0 kJ, the shooting frequency of the high-speed camera was 200,000 frame/s, and the total shooting duration was 895 μs. The detailed experimental conditions that related to the charging voltage, the in situ stress, and the energy in each specimen were shown in Table 2.

**Table 2.** Experimental scheme.

| Specimen No | In Situ Stress MPa | SSR | Frequency Frame/s | Acquisition Time µs | Explosive Energy kJ |
|---|---|---|---|---|---|
| S1 | 0 | 0 | | | |
| S2 | 8 | 0.1 | | | |
| S3 | 12 | 0.15 | 200,000 | 895 | 5 |
| S4 | 16 | 0.2 | | | |
| S5 | 24 | 0.3 | | | |
| S6 | 32 | 0.4 | | | |

## 3. Experimental Results

### 3.1. Evolution of the Crack Network without Static Stress

Figure 3 shows the initiation and propagation process of the crack network on the free surface of the specimen during crater blasting without static stress. Under the conditions using only the blasting dynamic load, four radial cracks, which are defined as the initial radial cracks (see Figure 3b), were first observed at the centre of the blast hole at 50 µs. At 80 µs, a first circumferential crack, which is defined as the initial circumferential crack (see Figure 3c), began to appear on the left of the blast hole.

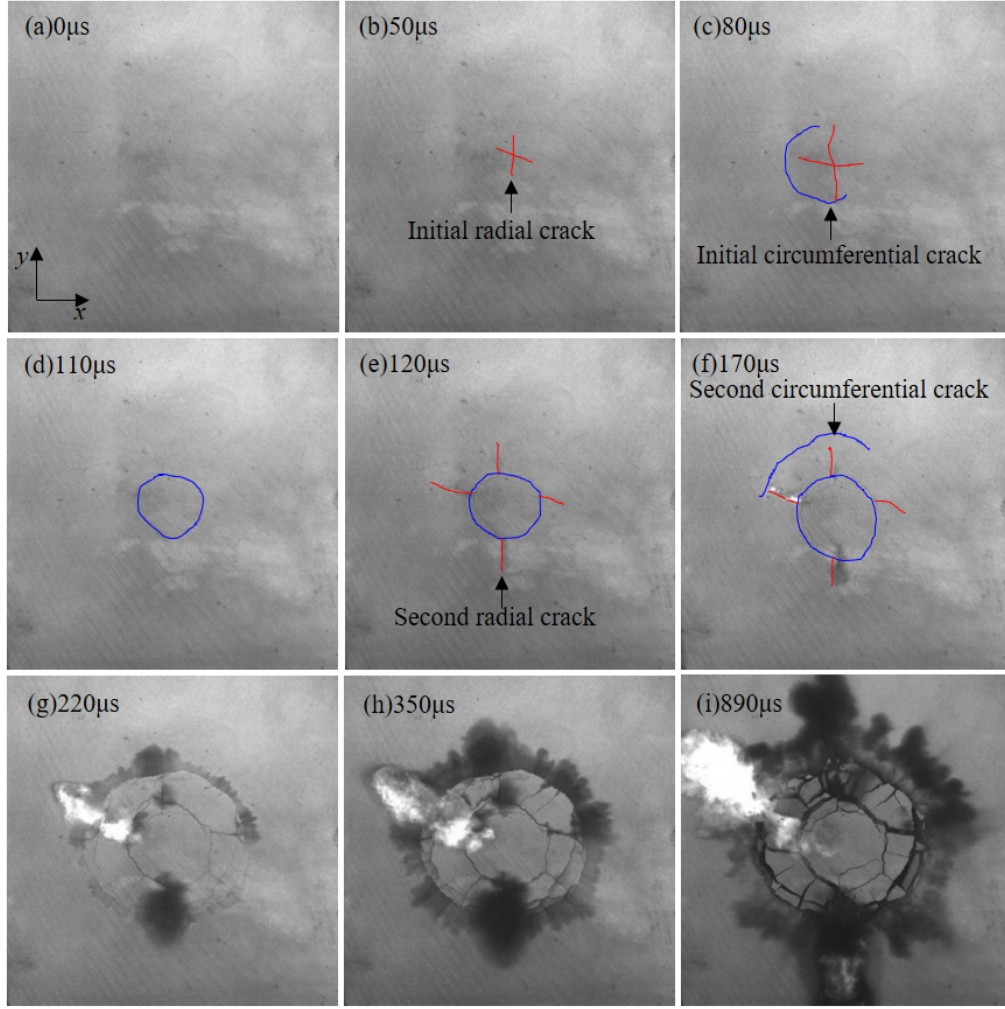

**Figure 3.** The crack network at different times during crater blasting under conditions without static stress. (**a–i**) The crack network after 0, 50, 80, 110, 120, 170, 220, 350, and 890 µs of explosion, respectively.

At 110 μs, the circumferential crack was completely formed (see Figure 3d). At 120 μs, during the formation of the first circumferential crack, the first observed radial cracks continued to propagate outward, which are defined as the second radial cracks (see Figure 3e). At 170 μs, a second circumferential crack appeared above the blast hole, and a small amount of blasting dust began to escape (see Figure 3f). At 220 μs, the main radial cracks and circumferential cracks were completely formed and the second circumferential crack was the boundary of the blasting crater. Over time, the blasting fragments moved outward, but new cracks no longer appeared. At 890 μs, a large amount of blasting dust was emitted, and the strong light generated by the electrical explosion was visible.

### 3.2. Evolution of the Crack Network under an SSR of 0.1

Figure 4 shows the initiation and propagation process of the crack network on the free surface during crater blasting under an SSR of 0.1. In Figure 4b, under the coupled dynamic and static loads, four radial cracks were observed first at the centre of the blast hole at 60 μs. At 110 μs, the initial circumferential crack appeared on the upper right side of the blast hole (see Figure 4c). At 140 μs, the first circumferential crack was completely formed and the secondary radial cracks could be observed outside the initial circumferential crack (see Figure 4d). At 150 μs, the second circumferential crack appeared on the left of the blast hole (see Figure 4e). At 270 μs, the main radial and circumferential cracks were fully formed, and blasting dust could be seen (see Figure 4f). As time passed, the blasting fragments flew outwards in all directions.

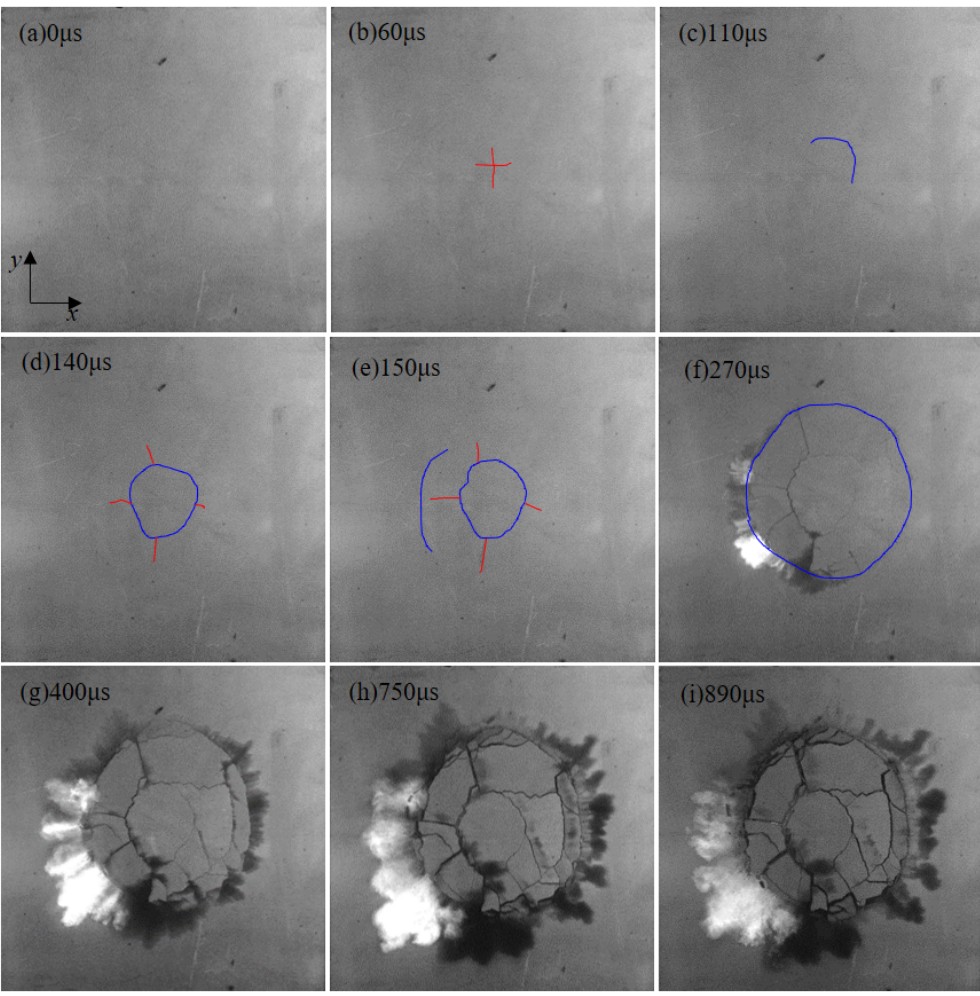

**Figure 4.** The crack network at different times during crater blasting under an of 0.1. (**a–i**) The crack network after 0, 60, 110, 140, 150, 270, 400, 750, and 890 μs of explosion, respectively.

### 3.3. Evolution of the Crack Network under an SSR of 0.15

Figure 5 shows the initiation and propagation process of crack network during crater blasting under an SSR of 0.15. Under the coupling of dynamic and static loads, four initial radial cracks were observed first at the centre of the blast hole at 70 µs (see Figure 5b). However, when compared with the radial cracks without static stress, the lengths of the four initial radial cracks under SSR of 0.15 were different. The radial cracks parallel to the static stress were longer than the radial cracks that were perpendicular to the static stress. At 110 µs, the initial circumferential crack started to appear on the right of the blast hole, as shown in Figure 5c. At 140 µs, the initial circumferential crack was completely formed (see Figure 5d). It was obvious that the initial circumferential crack was elliptical with its long axis parallel to the static stress. At 150 µs, the second radial cracks appeared outside the initial circumferential crack (see Figure 5e). At 170 µs, the second circumferential cracks started to appear on the left and right sides of the blast hole (see Figure 5f). At 250 µs, the second circumferential crack was fully formed, which stopped the propagation of the second radial cracks, and a small amount of blasting dust was emitted (see Figure 5g). Clearly, the second circumferential crack was also elliptical with a long axis that was parallel to the static stress. At 500 µs, the main radial and circumferential cracks were fully formed, and the blasting crater morphology was determined (see Figure 5h). After 500 µs, the blast fragments flew outward.

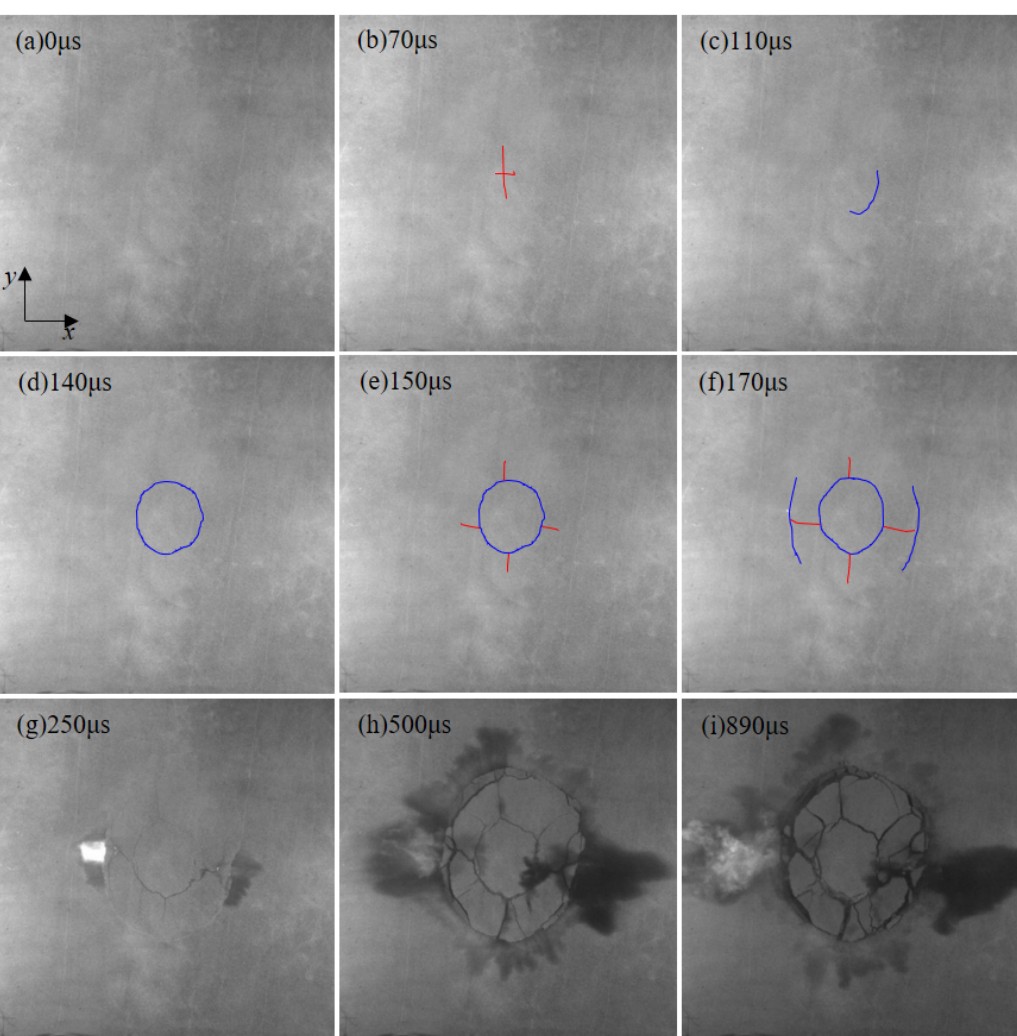

**Figure 5.** The crack network at different times during crater blasting under an SSR of 0.15. (**a**–**i**) The crack network after 0, 70, 110, 140, 150, 170, 250, 500, and 890 µs of explosion, respectively.

### 3.4. Evolution of the Crack Network under an SSR of 0.2

Figure 6 shows the initiation and propagation process of the crack network during crater blasting under an SSR of 0.2. Under the coupling of dynamic and static loads at 80 μs, a radial crack parallel to the static stress was observed first at the centre of the blast hole, while the radial cracks perpendicular to the static stress were not observed, as shown in Figure 6b. At 110 μs, the initial circumferential crack started to appear on the left of the blast hole (see Figure 6c). At 130 μs, the initial circumferential crack was completely formed. At the same time, two second radial cracks parallel to static stress direction were observed above and below the initial circumferential crack, as shown in Figure 6d. However, there were no second radial cracks perpendicular to the static stress outside the initial circumferential crack. At 160 μs, the second circumferential cracks, which tended to propagate along the static stress, started to form on both sides of the blast hole in the direction of the perpendicular static stress (see Figure 6e). At 200 μs, the second circumferential crack was closed, and the second radial cracks were randomly distributed between the initial and the second circumferential crack (see Figure 6f). At 300 μs, the main radial cracks and circumferential cracks were fully formed, when a small amount of blasting dusting started to leak out and the blasting crater morphology was determined (see Figure 6g). After 300 μs, the blasting fragments began to fly outward increasingly as time passed.

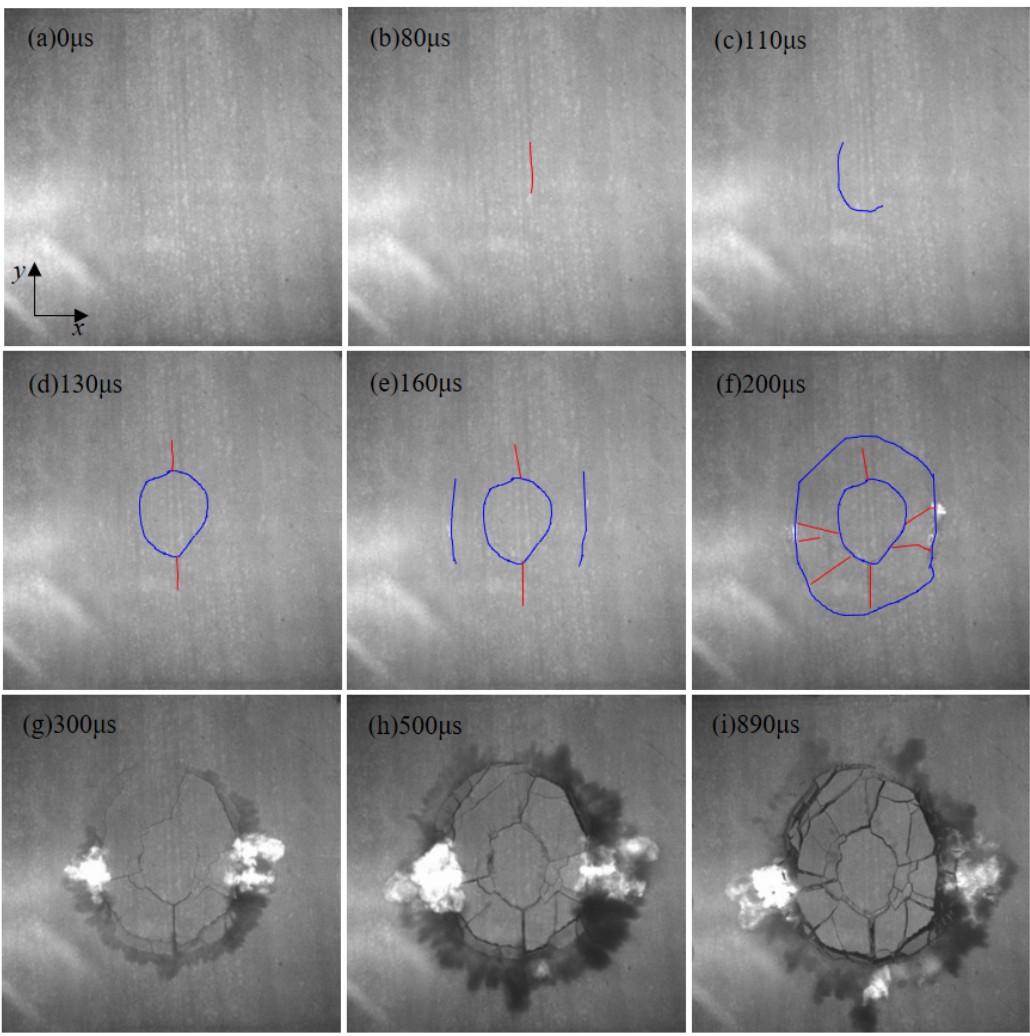

**Figure 6.** The crack network at different times during crater blasting under an SSR of 0.2. (**a**–**i**) The crack network after 0, 80, 110, 130, 160, 200, 300, 500, and 890 μs of explosion, respectively.

The morphological characteristics of the crack network under SSR of 0.2 were basically similar to those under an SSR of 0.15, both of which were elliptical with a long axis that was parallel to the static stress, as shown in Figures 5 and 6. However, when the SSR was 0.2, the radial cracks in the centre of the blast hole that were perpendicular to the static stress were completely restrained and did not appear. Moreover, when compared with the SSR of 0.15, the ellipse phenomenon of the crack network under the SSR of 0.2 was more obvious.

### 3.5. Evolution of the Crack Network under an SSR of 0.3

Figure 7 shows the initiation and propagation process of the crack network during crater blasting under an SSR of 0.3. Under the coupling of dynamic and static loads, an initial radial crack that was parallel to the static stress was observed first at the center of the blast hole at 80 μs (see Figure 7b). Similar to the experimental results that were obtained under the SSR of 0.2, radial cracks that were perpendicular to the static stress were not observed. At 120 μs, the initial circumferential cracks started to appear on both sides of the blast hole, perpendicular to the static stress (see Figure 7c). At 130 μs, the initial circumferential crack was completely formed. Meanwhile, the second radial cracks were randomly distributed outside the initial circumferential crack (see Figure 7d). At 170 μs, a second circumferential crack started to appear on the left of the blast hole, accompanied by a small amount of blasting dust that was starting to escape as well as light that was generated by the electrical wire explosion (see Figure 7e). At 300 μs, the second circumferential crack was closed (see Figure 7f). At 350 μs, the main radial and circumferential cracks were formed (see Figure 7g). After 350 μs, many small radial cracks outside the first circumferential crack gradually increased, and the blasting fragments flew outward.

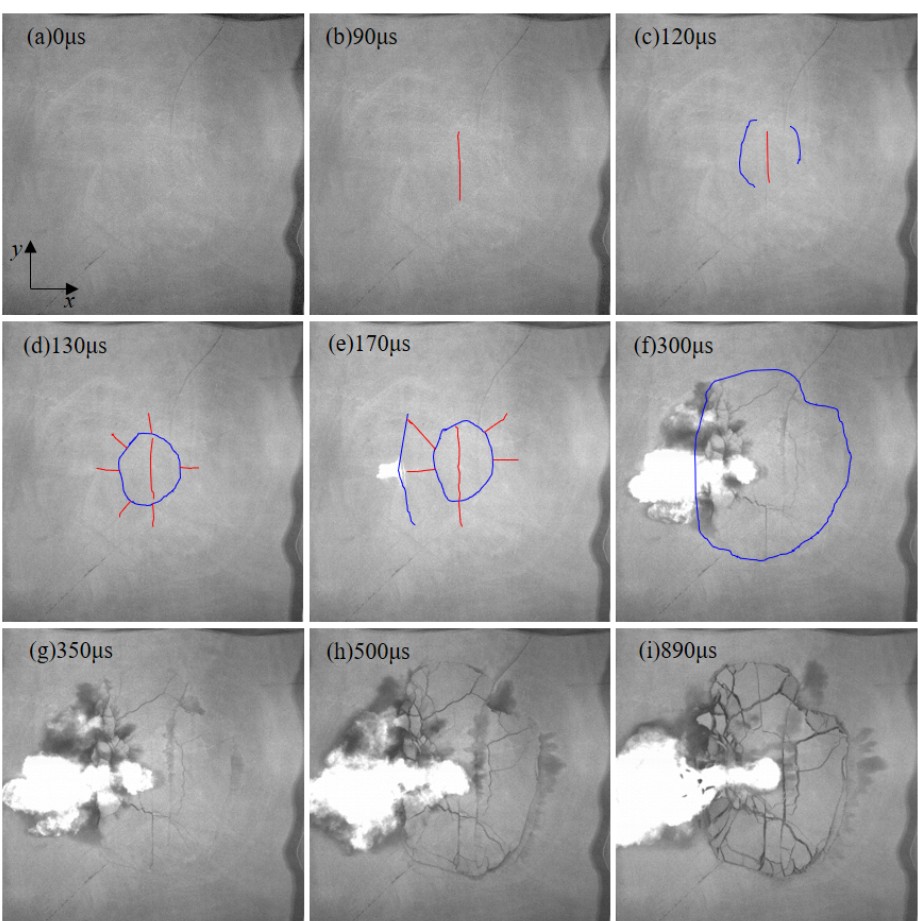

**Figure 7.** The crack network at different times during crater blasting under an SSR of 0.3. (**a–i**) The crack network after 0, 90, 120, 130, 170, 300, 350, 500, and 890 μs of explosion, respectively.

### 3.6. Evolution of the Crack Network under an SSR of 0.4

Figure 8 shows the initiation and propagation process of the crack network during crater blasting under an SSR of 0.4. Under the coupling of dynamic and static loads, similar to the experimental results of under an SSR of 0.2 and 0.3, only an initial radial crack that was parallel to the static stress was observed first at the centre of the blast hole at 150 μs (see Figure 7b). At 240 μs, the initial circumferential cracks started to appear on the left and right sides of the blast hole, accompanied by a small amount of blasting dust that was starting to leak out (see Figure 7c). At 300 μs, the second circumferential cracks also appeared on the left and right sides of the blast hole (see Figure 7d). Due to the effect of static stress, the initial and second circumferential cracks continued to propagate along the direction of the static stress. At 400 μs, the initial circumferential crack gradually closed. At the same time, two radial cracks appeared outside the initial circumferential cracks that were parallel the static stress (see Figure 7e). After 500 μs, the second circumferential crack started to bend and close and the blasting fragments began to fly outwards.

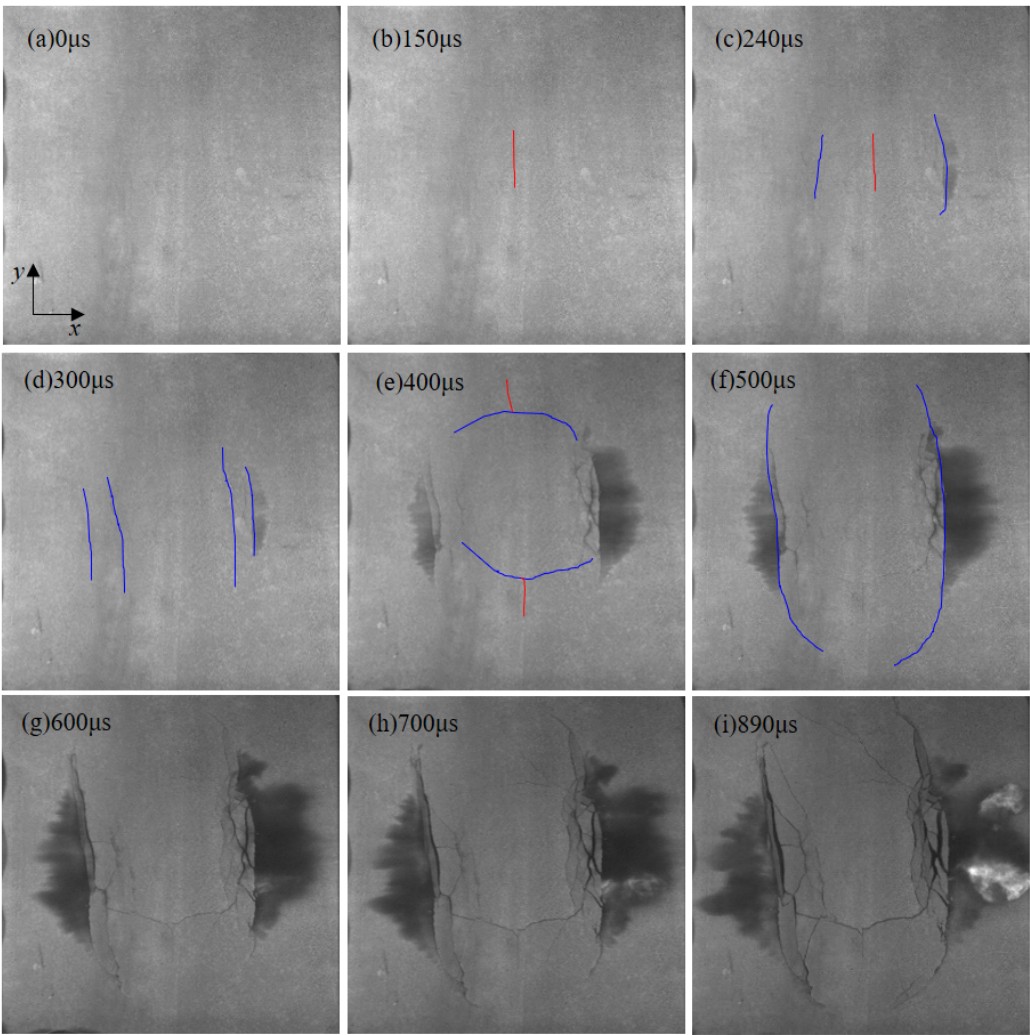

**Figure 8.** The crack network at different times during crater blasting under an SSR of 0.4. (**a–i**) The crack network after 0, 150, 240, 300, 400, 500, 600, 700, and 890 μs of explosion, respectively.

## 4. Discussion

### 4.1. Effects of Static Stress on the Formation Time and the Type of Initial Crack

Figure 9 shows the formation time and type of initial cracks under uniaxial static compressive stress. The formation time of the initial cracks without static stress condition was 50 μs after explosion. With the increase of uniaxial static stress, the formation time

of initial cracks were delayed. When the SSR reached 0.4, the time of the initial cracks' appearance suddenly was delayed even more, with the time of the first observed cracks being after 150 μs. The abovementioned experimental results indicate that static stresses tend to cause a time delay of the cracks' appearance. Therefore, the effects of static stress on the formation time of cracks should be considered during blasting excavation of deep, high geo-stress rock engineering. In other words, the appropriate millisecond blasting time should be selected according to the in situ stress, and this would be helpful in improving the blasting effect.

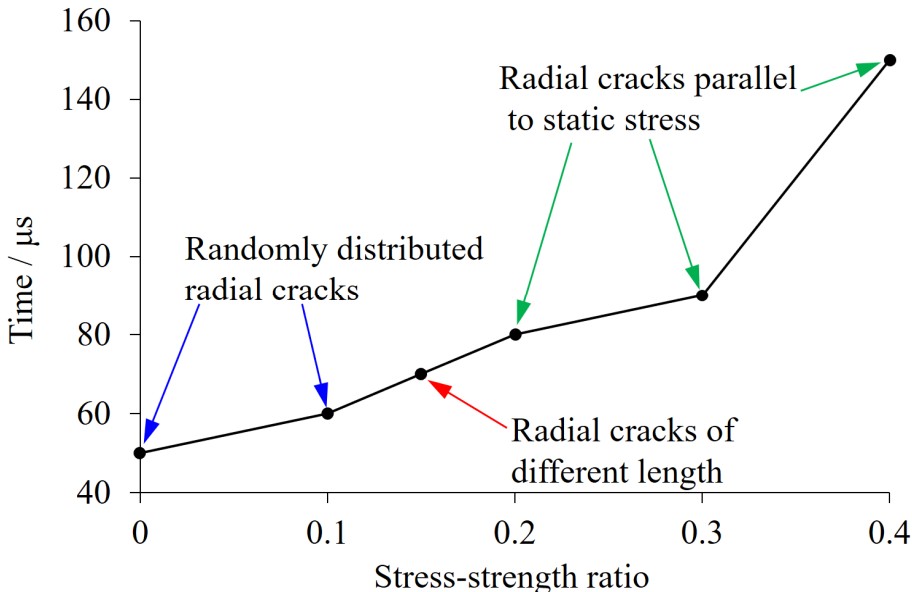

**Figure 9.** The formation time and type of initial cracks under uniaxial static stress.

Furthermore, the uniaxial static stress also changes the type of initial cracks as is shown in Figure 9. When the SSR is 0 or 0.1, the initial cracks are randomly distributed radial cracks. When the SSR is 0.15, the radial cracks that are parallel to the static stress are longer than the radial cracks that are perpendicular to the static stress. However, when the SSR reaches 0.2 and above, the only initial cracks are those radial cracks that are parallel to the static stress, and the radial cracks perpendicular to the static stress are inhibited by the static stress and do not appear. With the increase of uniaxial static stress, the type of initial cracks also changes. The above experimental results show that the static stress promotes the initiation and propagation of radial cracks that are parallel to its own direction and inhibits the initiation and propagation of radial cracks that are perpendicular to its own direction.

It should be noted that the mechanism of the delay in the appearance of initial cracks under uniaxial static stress needs to be further studied, which would be helpful for the reasonable selection of the delay time during millisecond blasting.

### 4.2. Effects of Static Stress on the Morphological Characteristics of the Crack Network

As shown in Figures 3 and 4, the initiation and propagation process of the crack network under the SSR of 0.1, and their morphological characteristics, are similar to those without static stress. Meanwhile, the crack network under the SSR of 0.1 is nearly circular and does not exhibit anisotropic characteristics. The above results suggest that when the SSR is 0.1 or below, the static stress does not have a significant impact on the crack network. Therefore, when the SSR ≤ 0.1, the effects of static stress on the propagation and morphology of the crack network are negligible.

When the SSR is 0.15, four radial cracks form at the centre of the blast hole, with longer radial cracks parallel to the static stress and shorter radial cracks that are perpendicular to the static stress. By comparing Figure 3 with Figure 5, it can be seen that the static stress

causes the crack network to be elliptical with a long axis that is parallel to the static stress, which indicates that when the SSR reaches 0.15, the static stress has a non-ignorable effect on the propagation and morphology of the crack network. Therefore, when the SSR reaches 0.15 and above, it is necessary to consider the effects of static stress on the crack network in rock blasting.

According to Figures 6–8, the anisotropy of the crack network gradually increases with the increase of the SSR. Therefore, the anisotropy of the crack network is positively correlated with the anisotropy of static stress. Yi et al. [29] and Peng et al. [30] also obtained the same experimental results by a serious of experiments. The above experimental results indicate that the blast hole spacing should be properly adjusted based on the in situ stress.

### 4.3. Relationship between the Strain Field Morphology and the Crack Network Morphology

Figure 10 shows the maximum principal strain field prior to the appearance of cracks and the crack network during crater blasting [24,25]. The maximum principal strain field was obtained by a model experiment, where the experimental method that was used in this study as well as a digital imagine correlation (DIC) technique were used. In this study, the tensile strain is positive and the compressive strain is negative. Clearly, the morphological characteristics of the maximum principal strain field and the crack network are similar, both of which are ellipses with the long axis parallel to the uniaxial static stress. The similarities of the morphological characteristics between the strain field and the crack network indicate that the strain field prior to the cracks' formation during rock blasting controls the initiation and propagation process of the crack network, and ultimately its morphological characteristics.

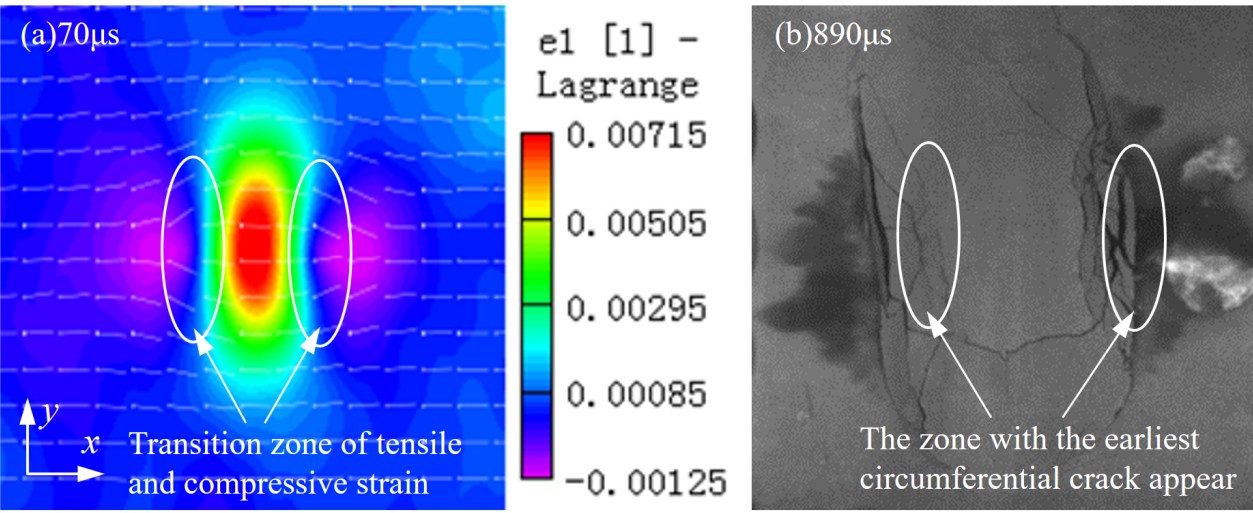

**Figure 10.** Maximum principal strain field and the crack network. (**a**) The maximum principal strain field under an SSR of 0.42 after 70 μs of explosion; (**b**) the crack network under an SSR of 0.4 after 890 μs of explosion.

In Figures 6–8, the circumferential cracks always appear on both sides of the blast hole perpendicular to the static stress. As shown in Figure 10a, the zones of both sides of the blast hole that are perpendicular to the static stress are the transition zone of the tensile and compressive strain. Therefore, the circumferential cracks preferentially appear in the transition zone of the tensile and compressive strain, and propagate along the direction of the static stress. Moreover, blasting dust and light that were generated by the electrical explosion always escaped first from the transition zone (Figures 6–8), which suggests that under uniaxial static stress, the rocks on both sides of the blast hole perpendicular to the static stress are more prone to fracture.

In the transition zone of tensile and compressive strain, the shear strain resulting from tensile and compressive strain causes the shear failure of the rock [45,46]. Therefore, the

shear cracks will occur between the bottom of the blast hole and the free surface at the transition zone of the tensile and compressive strain. Then, the blasting dust and the light that is generated by electrical explosion first escape from the transition zone of the tensile and the compressive strain.

### 4.4. Mechanism of the Uniaxial Static Stress Affecting Crack

The tensile fracture is the main failure mode of rock under a dynamic load. According to the maximum tensile strain failure criterion, rock will fracture if the strain that is caused by explosive dynamic load reaches the dynamic ultimate tensile strain of rock, as shown in Equation (1) [47,48].

$$\varepsilon_{ds} \geq \varepsilon_{td} \tag{1}$$

where: $\varepsilon_{ds}$ is the tensile strain caused by the coupling of dynamic load and static loads and $\varepsilon_{td}$ is the dynamic ultimate tensile strain of rock.

The static strain field caused by static stress of $y$-axis is:

$$\varepsilon_1 = \varepsilon_{sx} = \frac{\mu \sigma_y}{E}$$
$$\varepsilon_2 = \varepsilon_{sy} = -\frac{\sigma_y}{E} \tag{2}$$

where: $\varepsilon_1$ and $\varepsilon_2$ are the maximum and minimum principal strains on the free surface of the specimen, respectively; $\varepsilon_{sx}$ and $\varepsilon_{sy}$ are the static strains in the $x$-axis and $y$-axis direction that are induced by static stress of $y$-axis, respectively.

According to Equation (2), under the static stress of the $y$-axis, the static strain field in the $x$-axis and $y$-axis direction are tensile strain and compressive strain, respectively.

Figure 11 shows the maximum and minimum principal strain field on the free surface that are caused by $y$-axis static stress [24], which were obtained by the model crater blasting experimental method that was used in this study and a DIC. Clearly, the maximum principal strain field is positive, which is the tensile strain field, whose direction is perpendicular to the static stress while the minimum principal strain field is negative, which is the compressive strain field, whose direction is parallel to the static stress. The experimental results presented in Figure 11, are consistent with the theoretical analysis results in Equation (2). Meanwhile, the direction of the maximum principal strain field is always perpendicular to the static stress during crater blasting (see Figure 10a).

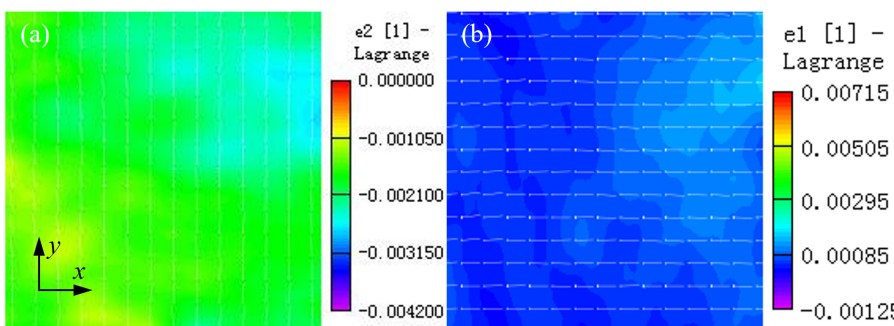

**Figure 11.** The static strain field that is induced by static stress of $y$-axis. (**a**) The minimum principal strain field; (**b**) the maximum principal strain field.

Due to the effects of the strain field that is induced by uniaxial static stress, the cracks inevitably propagate along the $y$-axis direction which eventually leads to an elliptical crack network. That is why the crack network is an ellipse with its long axis that is parallel to the static stress under uniaxial static stress.

## 5. Conclusions

In this paper, the effects of anisotropic in situ stress field on crack initiation and propagation during hard rock blasting were investigated using crater blasting experiments of green sandstone under various uniaxial static stresses. The main conclusions are as follows.

(1) Uniaxial static stress changes the type of initial cracks during rock blasting and leads to the delay of the initial cracks' formation and propagation.
(2) Under uniaxial static stress, circumferential cracks are prone to appear on both sides of the blast hole, perpendicular to the static stress and propagate along the static stress.
(3) Under uniaxial static stress, the crack network during rock blasting is an ellipse with its long axis parallel to the static stress. The anisotropy of the crack network is positively correlated with the anisotropy of the static stress.
(4) The strain field prior to crack appearance controls the initiation and propagation of cracks and determines the morphological characteristics of the crack network.

**Author Contributions:** Conceptualization, G.Y. and F.Z.; methodology, G.Y. and F.Z.; investigation, G.Y. and Q.Y.; experiments, G.Y., Q.H., Q.Y., X.W. and H.W.; data curation, G.Y. and Q.H.; Writing—Original draft preparation, G.Y.; writing—review and editing, F.Z.; supervision, F.Z.; project administration, F.Z.; funding acquisition, F.Z. All authors have read and agreed to the published version of the manuscript.

**Funding:** This work was supported by the State Key Research Development Program of China (grant no. 2017YFC0602902).

**Institutional Review Board Statement:** Not applicable.

**Informed Consent Statement:** Not applicable.

**Data Availability Statement:** Not applicable.

**Acknowledgments:** The first author thanks the China Scholarship Council (No. 202008060081) for support, which enabled him to study in the School of Engineering, National University of Singapore.

**Conflicts of Interest:** The authors declare that they have no conflict of interests and no known competing financial interests or personal relationships that could have appeared to influence the work reported in this paper.

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
