# Peer review of "Experimental Study on the Effects of In Situ Stress on the Initiation and Propagation of Cracks during Hard Rock Blasting"

_applsci, doi:10.3390/app112311169_

Round 1

Reviewer 1 Report

The paper concerns the results of an experimental tests campaign reproducing the underground mining method called Vertical Crater Retreat. The lateral confinement due to the lateral earth pressure is represented by an uniform unidirectional stress state performed by means of a compressive testing machine. Different increasing static stress states are investigated. The theme is attractive and actual and can be considered suitable for the journal. Since experimental data are not easily available, the results of laboratory tests are always welcome. The attention of this referee has been given mainly to the central parts of the paper (corresponding to paragraph 3 and 4): experimental test campaign and analytical interpretation of the results. These main aspects have been examined in detail, since it is opinion of this referee that a corrections are needed in order to improve both reading and utility. Moreover, a general revision by a native English speaker is suggested.

Paragraph 3. Experiments

The usefulness of the results of laboratory test campaign is, like all experimental data, strongly linked to the possibility of every reader to follow step by step the analysis performed. The authors should clarify in every aspect the performed experimental campaign. There are some unsolved questions:

  1. The only difference with the Vertical Crater Retreat is the hole direction, that in the experimental tests is orthogonal to the real cases. Can the author explain how this different condition affects the in situ explosions?
  2. The lateral earth pressure, even in the mining real conditions is multiaxial, what is the utility of analyzing uniaxial loadings?
  3. What is the uniaxial confinement stress direction? In figure 1 a testing machine able to perform biaxial compressive tests (two actuators are clearly visible). In the text a clear indication of the load axis (is it x or y, see Fig. 2?) is necessary. The paper would benefit if a graphical indication of the load axis is included in the specimen representation of figure 2.
  4. The authors should specify if the mechanical characteristics of the sandstone have been detected by means of experimental tests performed by the authors or are derived from literature. In both cases a reference should be provided.

The authors should address points a) and b) in the text and summarize the main points in the conclusions

Paragraph 4. Discussion

In this part too there are some clarifications to do:

  • Large part of the discussion involves comparisons of the experimental crack maps with those derived from numerical analyses (i.e. Figures 10 (a) and 11). How the strain maps were obtained? Are they derived from a finite element analysis? Is it a commercial code or a routine developed by the authors? What is the adopted constitutive model? How were the cracks taken into account with the evolution of the crack map? In other words, since the appearance of the first crack, what are the changes done at the constitutive model or at the geometry of the sample?
  • Lines 265-366, the statement “the crack network is ellipse with the long axis parallel to the static stress under uniaxial static stress” seem uncorrect. Please check the position of the major axis of the ellipse with respect to the stress direction.

Other minor corrections

Figure 2: Please correct picture (b) caption. Moreover,a rotation of the picture to make it corresponding to the photo in (c), could be clarifyng

Lines 264-274 The symbol ms is uncorrect, it is perhaps a typo. Please check all along the paper

Reviewer 2 Report

Language

The language of the manuscript is good but can be improved for a final publication.

Technical

  1. The Introduction part discusses and presents numerous peripheral references to the topic at hand, such as the status of in-situ stresses, its variation with depth, and rock properties. However, very little is mentioned regarding the basics of fragmentation under high in-situ stress conditions, as well as other studies that have examined this topic. The reference by Yi et al (2018) has almost exactly studied the same phenomenon but it is mentioned in the Results in section 4.2 (p. 12).
  2. Strangely, no examples or data are given from underground metal mines in any country, even though they have been conducting blasting for developments and production on a daily basis for more than 100 years. The topic is presented in such a manner that the reader might think that no one is currently able to drill-and-blast rock in high stress environments.
  3. The experimental results take up 7 of the 16 pages but they present the same results for 6 different SSR values. The descriptions of the results are similar with slight variations in the photos, of course. These could have been summarized in a table with only the final 3 photos of each SSR being presented for comparison.
  4. At no time is it indicated that a numerical approach is also being used for the study. The reader learns this for the first time in section 4.3 on p. 13. Moreover, the model results are from two other publications by the authors that had not been mentioned until this point. The use of numerical models is introduced for the first time in section 4.2 on p. 12 where two new references are mentioned that had not been in the Introduction.
  5. No relationship is established between the study conclusions and field results or usage, such as at an actual mine or power station. The case studies first mentioned in the Introduction are not brought up again or compared to the results obtained.
  6. The study is needlessly simplistic and looks into a single blasthole and its effect on crack propagation in a rock panel. Such studies have been covered in basic blasting textbooks and need not be repeated here. In the field, blasting is not conducted with a single hole but a network of holes. It would have been very interesting to see how cracks formed and interacted between various blastholes with multiple patterns rather than a single hole.

The manuscript needs to take into account all these points and fully address them before being considered for publication.

Reviewer 3 Report

Dear Authors,

you made experiments related to propagation of cracks during hard rock blasting and your presentation in a form of manuscript was submitted to Applied Sciences. Of course the subject is important in many aspects but the fundamental problem is to indicate scientificity in the presented approach. Is the paper is a technical note or article with implications defined by editors as “the work reports scientifically sound experiments”. In my opinion “scientifically sound” is just a scientific standard which in this case is different than a report of laboratory tests.

For scientific papers, reproducibility and replicability of the scientific results are fundamental concepts that allow other scientists to check and reproduce the results under the same conditions described in the paper or at least similar conditions and produce similar results with similar measurements.

In old times of my master dissertation I had to spend months to make hundreds tests to obtain statistical regularity to formulate some conclusions (cracking in rocks subject to thermal stresses). The authors formulate general conclusions but they don’t mention that they are true only for the specimen. OK – some results were confirmed by other authors mentioned in the text. How the authors know that other specimens will provide the same results? All statements are true only in these individual cases, therefore they are not universal and do not bring any knowledge about the processes under study.

Unfortunately, the work is not of a scientific nature and does not meet the aims and scopes declared by editors of the journal. What is a scientific aim here or what is to be applied?

The authors conducted interesting laboratory experiments concerning the analysis of the fracture process.

Unfortunately, the paper does not provide any theoretical foundations related to energy release and modeling of the crack propagation process that would allow for example a comparative analysis as comparison between of simulated crack pattern and experimental crack pattern.

There are no references to literature, including basic works. This leads to a false perception of this work as pioneering in the studied area. It is a very unfair practice, when it comes to academic work. It is different in the case of the technical report on the performed experiment. Anyway, this is how I perceive the work done as a technical report with hastily made drawings, where the results were obtained and there are also conclusions, but they are not of general importance and concern specific samples and specific conditions. Even in this case, no parameters of rocks are presented, only parameters for the electrical explosion and the camera which was used to collect the failure process.

Many studies have been done in the field of rock crack initiation and propagation induced by blasting under preexisting stress and they present a good scientific scheme of manuscript

  1. A. Persson, N. Lundborg, and C. H. Johansson, “The basic mechanisms in rock blasting,” International Society of Rock Mechanics Proceedings, vol. 3, pp. 19–33, 1970
  2. K. Kutter and C. Fairhurst, “On the fracture process in blasting,” International Journal of Rock Mechanics and Mining Sciences and Geomechanics Abstracts, vol. 8, no. 3, 1971.
  3. McHugh, “Crack extension caused by internal gas pressure compared with extension caused by tensile stress,” International Journal of Fracture, vol. 21, no. 3, pp. 163–176, 1983.

You have to place your work in large area of many studies devoted to the problem.

“Things have their roots and branches. Affairs have their beginnings and their ends. To know what is first and what is last will lead one near the Way.”

"Yi," "Zhi," and "Xin" are perfectly applicable to scientific research and scientific texts!

Dear authors! Please do not get discouraged or depressed!

Many important parts are missing in the current version of your manuscript and my stringent comments above aim to improve your work and a stronger paper for the future.

I am passing detailed corrections of the text you will find in the attachment. I am giving you as a gesture of goodwill and I hope that will be helpful!

Round 2

Reviewer 2 Report

Significant improvements to the text have been made. Multi-borehole studies can be left for another study in the future.

Reviewer 3 Report

Dear authors, it is with great satisfaction that I read your replies to the remarks. 
Now your work looks much better, so you see "nothing in the world is worth having or worth doing unless it means effort, pain, difficulty".

Have satisfaction with fruits of your hard labour, which effect will be published soon (I hope).

:)